# Feline Encounters Down Under: Investigating the Activity of Cats and Native Wildlife at Sydney’s North Head

**DOI:** 10.3390/ani14172485

**Published:** 2024-08-27

**Authors:** Brooke P. A. Kennedy, Anna Clemann, Gemma C. Ma

**Affiliations:** 1School of Environmental and Rural Science, University of New England, Armidale, NSW 2351, Australia; bkenne27@une.edu.au (B.P.A.K.); aclemann@myune.edu.au (A.C.); 2Royal Society for the Prevention of Cruelty to Animals New South Wales, Yagoona, NSW 2199, Australia; 3Sydney School of Veterinary Science, The University of Sydney, Camperdown, NSW 2006, Australia

**Keywords:** *Felis catus*, free-roaming, containment, camera trap, wildlife, conservation

## Abstract

**Simple Summary:**

Cats are a common sight across Australia, freely roaming not only through urban areas but also within natural habitats, including protected areas like national parks. This unrestricted movement raises concerns due to potential impacts on native wildlife populations. North Head, Manly, located in New South Wales, serves as a prime example, boasting a rich biodiversity that includes endangered populations of Long-nosed Bandicoots and Little Penguins. Recent observations by wildlife officers within Sydney Harbour National Park and the North Head Sanctuary highlighted the presence of cats, prompting a deeper investigation into their spatial and temporal distribution. Cameras were installed across the headland over a five-week period to capture cat and native fauna activity. Cats were frequently observed at the interface with the urban area of Manly. Moreover, cat activity primarily occurred during the night, coinciding with periods of heightened native mammal activity. These findings underscore the potential for direct and indirect interactions between cats and native wildlife within the headland, with implications for species conservation efforts. This study emphasises the importance of implementing proactive management strategies to mitigate the potential impact of feline predation on local biodiversity while also highlighting the need for further research in this area.

**Abstract:**

Cats (*Felis catus*) are widespread across Australia, including within natural and protected areas, and in many areas, cats, including owned domestic cats, are not restricted in where or when they can roam. In Australia, cats have contributed to the decline of many native species and continue to be a problem for governments. North Head, Manly, is home to an endangered population of Long-nosed Bandicoot (*Perameles nasuta*) and the only mainland breeding colony of Little Penguin (*Eudyptula minor*) in New South Wales (NSW). Camera traps were installed for a 5-week period across North Head to determine the spatial and temporal distribution of cat activity. As well as capturing instances of cats, the cameras detected native animals such as birds, possums, Long-nosed Bandicoots and other small mammals. An analysis of the camera images showed cats could be found within protected areas of the headland (where cats are prohibited) and along the boundary with the adjacent suburban area of Manly. Cats were mostly detected during the night. There were high occurrences of overlap between cats and Long-nosed Bandicoots (Dhat 0.82), possums (Dhat 0.88) and other small mammals (Dhat 0.67). These findings indicate that cats are active across the Manly headland at the same time as native animals, both within protected areas where cats are prohibited and in adjacent residential areas, and this could have implications for these populations.

## 1. Introduction

Cats are popular companion animals in Australia and are also widespread across the continent, including within natural and protected areas. Australian wildlife, particularly small mammals, are uniquely vulnerable to cat predation. Cats are thought to have contributed to more than twenty Australian mammal extinctions since European colonization [1], and predation by cats is recognized as a key threatening process with a significant detrimental impact on many extant threatened species [2]. Roaming cats also create a community nuisance through urine and fecal soiling, noise and property damage while also being at substantial risk of accidents and injury [3]. At present, in the state of New South Wales (NSW), cats are prohibited from entering national parks and areas declared ‘Wildlife Protection Areas’ (WPAs) by local councils but are otherwise allowed to roam without restrictions.

Cats (*Felis catus*) were introduced to the Australian mainland during European colonization in the late 1700s [4] and have since spread across mainland Australia and to many islands [5]. The cat population in Australia can be divided into three important sub-populations: feral cats, which are not reliant on humans for food or shelter and live and reproduce in wild areas (such as forests, wetlands, deserts, etc.); unowned domestic cats can be found in urban, semi-urban and rural areas and may obtain some food and shelter (knowingly or unknowingly) from humans, but are not owned by them; owned domestic cats are knowingly owned and cared for by humans and are provided with food and shelter on a regular basis [3]. The exact size of the cat population in Australia, including the size of the owned and unowned domestic and feral sub-populations, is estimated to be between 7 and 11.2 million, including 4.9 million owned cats [6].

Given the ongoing decline of native wildlife in Australia, it is important to better understand the impact of cats in our natural areas. The Manly headland in NSW encompasses medium-density residential developments, the Sydney Harbour National Park (state government) and North Head Sanctuary (federal government), along with various other land tenures. North Head provides important habitat for small native mammals, including Bush Rats (*Rattus fuscipes*), Brown Antechinus (*Antechinus stuartii*) and the vulnerable Pygmy Possum (*Cercartetus nanus*), some of whom have been re-introduced to the area in recent years [7]. The headland is also home to an endangered population of Long-nosed Bandicoots (*Perameles nasuta*) [8], the eastern pygmy possum (*Cercartetus nanus*), listed as vulnerable, and an endangered Little Penguin (*Eudyptula minor*) population, which is the only penguin breeding site on mainland NSW [9]. Due to ongoing land clearing for expansion of urban development in Australia—particularly on the east coast—wildlife habitats are becoming increasingly fragmented [10]. Remnant fragments of habitat, such as that preserved at North Head, might be disproportionately impacted by domestic cat predation. While individual domestic cats are estimated to predate far fewer native animals than their feral counterparts, at a population level, predation pressure on native birds, mammals and reptiles exerted by domestic cats is estimated to be 30–50 times greater per square kilometer than predation by feral cats due to their greater population density [11]. While cat roaming behavior varies considerably, several studies in different settings have suggested that cats living adjacent to natural areas roam further and prefer roaming within natural areas [12,13,14,15]. As such, domestic cats are suspected to create a predation ‘halo effect’ where prey populations are reduced in the areas surrounding urban development. However, direct evidence of this from Australia is lacking.

The headland at Manly (hereafter referred to as North Head) is a tied island. Its history of housing a Quarantine Station and military facility has precluded urban development and preserved much of the natural environment. As a small remnant fragment of wildlife habitat surrounded by residential development, North Head offers an important case study for both endangered species conservation and domestic cat management. This study aimed to determine the spatial and temporary activity of cats across North Head. This study also aimed to define which subpopulation of cats (feral, owned or unowned domestic) access protected areas of North Head where cats are prohibited and determine which access points are used.

## 2. Materials and Methods

### 2.1. Setting

This study was conducted at North Head in NSW, Australia. The suburb of Manly has a population of approximately 83,208 people in 35,282 private dwellings with a density of approximately 5496 people per square kilometre (83,208/15.43 km^2^) [16].

North Head includes a variety of land tenures (Figure 1). Cats are prohibited within the Sydney Harbour National Park and North Head Sanctuary (managed by the Sydney Harbour Federation Trust), which are adjacent to the residential area of Manly and include sclerophyll forest and littoral rainforest remnants and coastal areas. There are no residential dwellings within the Sydney Water Treatment Plant. The coastal areas are made up of small shrubs and heath that lead into sandy beaches and tall cliffs on the east and south faces of the headland. The headland includes the largest extant occurrence of the Eastern Suburbs Banksia Scrub, a critically endangered ecological community [17,18,19].

General weather conditions for Manly are temperate, with a range of mean maximum temperatures between 17 °C and 25 °C and mean minimum temperatures between 10.5 °C and 20.5 °C. The rainfall ranges between 62 and 152 mm per month [20]. As seen in Table 1, the rainfall during the study period (May to July) was low for this area, and the minimum and maximum temperatures were average. 

### 2.2. Remote Sensing Cameras

A total of 19 heat-in-motion cameras (Swift 3C wide-angle, Outdoor Cameras Australia, Toowoomba, QLD) were deployed across North Head, including within the state and federal protected areas and along their northern boundary (Figure 1). Of 20 cameras, 10 were deployed along the residential boundary of the state and federal protected areas—including two within the boundaries of the Sydney Harbour National Park (cameras 4 and 5), five on adjacent Northern Beaches Council land (cameras 1, 2, 6, 8 and 9), two on adjacent privately owned land of the International College of Management (cameras 7 and 20) and one on the campus of St Paul’s Catholic College (camera 3)—and 9 were deployed deeper within the Sydney Harbour National Park (cameras 10 and 12) and the North Head Sanctuary (cameras 11, 13–16, 18 and 19) (faulty cameras were not deployed, camera 17). Cameras were deployed for a period of 56 days from late May until July 2023.

Cameras were deployed using settings optimized to capture predators, including cats and foxes [21]. Other target species that were expected to be captured using these settings included the Long-nosed Bandicoot (*Perameles nasuta*), Brushtail Possum (*Pseudocheirus peregrinus*), Ringtail Possum (*Trichosurus vulpecula*), birds, lizards, Echidna (*Tachyglossus aculeatus*) and other native and non-native small mammals.

Cameras were attached vertically to trees or established poles along or near walking tracks. Cameras were positioned at a height of 90 cm from the ground and were assisted to angle downward by a pre-cut piece of dowel. Cameras deployed along tracks were angled at approximately 22° to the tracks, whilst others were pointed directly at an object, i.e., where two tracks joined into one or directly at a water source. Some minor pruning of leaves and small branches blocking the camera view was conducted. No other major vegetation movements were made. The following camera settings were used: high passive infrared (PIR) sensitivity (with a known 20 m range); three photo captures per trigger; zero intervals between triggers; image size of eight megapixels. 

### 2.3. Data and Statistical Analysis

The images were downloaded, viewed individually and manually tagged using the image tagging program ExifPro 2.1 (Bad Kreuznach, Rheinland-Pfalz, Germany). If objects were identifiable, they were given one or two (if more than one object appeared in an image) of the following tags: ‘BAN’ (Long-nosed Bandicoot), ‘BIRD’, ‘CAT’, ‘DDOG’ (domestic dog, *Canis familiaris*), ‘FOX’ (*Vulpes vulpes*), ‘HUMN’ (human), ‘LIZ’ (lizard), ‘POS’ (possum), ‘RAB’ (rabbit), ‘ECH’ (echidna) or ‘SMLMAM’ (any other mammal smaller than a Long-nosed Bandicoot, including invasive mice and rats or bush rats and antechinus). If not identifiable, the images were tagged with ‘UNK’ (unknown: the image was too dark to identify and/or the object was blown out by the camera flash) or ‘NIL’ (nothing identifiable in the image). 

Tagged images were loaded into the R Studio program (2022.07.2 Build 576; Boston, MA, USA) in groups of three images per trigger. An event was defined as anything that was captured 60 s or longer apart from the previous event, and this counted as one observation. The events were counted, with any unidentifiable images and those with missing data removed. A clean dataset was compiled into a csv. file by camera number, date, time and tag. 

After a simple count of the data, the overlap package in R Studio was used to determine the coefficient of overlapping of ‘CAT’ and ‘BAN’, ‘POS’, SMLMAM’ and ‘BIRD’. The Dhat4 estimator was used due to the large sample sizes. 

Using all the images tagged ‘CAT’, individual cats were identified and monitored across all cameras. Cats were identified via their body markings by B.P.AK. Cats that were unable to be identified were listed as ‘Unknown’ and not counted as individuals in case the cat was captured elsewhere. Therefore, the number of individual cats presented is the minimum number of cats in the area at the time the research was conducted. Cats wearing collars or of identifiable breeds were also noted. The breed was determined by a visual appraisal of images by G.C.M. Occupancy modelling in R Studio was used to predict the occupancy and detection rates of cats within the national park.

An ANOVA statistical analysis was run to determine any difference between the number of cats seen on the different cameras.

## 3. Results

A total of 171,669 camera trap images captured 23,218 true events (Table 2) and 34,005 NIL or UNK events over 963 trap nights. Of the 23,218 events, 34% (n = 8134) were animals tagged under the 10 categories (Figure 2), and the remaining 66% (n = 15,268) were humans (Table 2). 

Eleven individual cats were captured in 50 events on 63% of the cameras (12/19; Table 3). One cat image, detected on camera 12, was too indistinct to identify individually. Of the 11 cats, 4 were visibly wearing collars and 1 was being walked on a lead by a human. Four cats were of identifiable breeds, including one Ragdoll, one Bengal and two Burmese. Cats 3, 5, 6 and 7 were recorded on two cameras, cat 4 was recorded on five cameras and the remaining cats were recorded on one camera each (Figure 3). Cats 4 and 7 were captured on two cameras on the same night. The spatial and temporal data for these six events (3 nights) are shown in Figure 4. Five individual cats, two of whom were wearing collars and were of an identifiable breed (Ragdoll and Burmese), and a third who was identified as a Burmese, were detected in 24 separate events by cameras located within state or federal protected areas where cats are prohibited (Cats 3–7; Table 3). All cats wearing collars and/or of identified breeds were considered owned domestic cats (6/11). The remaining five cats were all detected within 100 m of human homes within suburban Manly (cameras 1–4 and 20). Hence, these cats were also likely owned domestic cats; however, it cannot be ruled out that these cats were unowned domestic cats.

Two cameras (2 and 4) accounted for over half of the cat events (Figure 4). Camera 2 recorded the most cat events (15). This camera was located within a council reserve, approximately 100 m from the boundary of the Sydney Harbour National Park near the Collins Flat Track, a popular walking route through a section of littoral rainforest (Figure 5a). Camera 4 recorded the next most cat events (11) and was located along a walking track through coastal scrubland just inside the Sydney Harbour National Park boundary wall (Figure 5b). Although there were more cats captured on cameras located at the urban/natural edge, this was not significantly different from the number of cat events deeper within the protected areas (F_1,17_ = 1.527, *p* = 0.233).

Cats were recorded at or between dusk and dawn, with peaks in cat activity recorded around 6.00 p.m. and 2.00 a.m. and a small peak at 12.00 p.m. (Figure 6). This temporal cat activity was plotted against wildlife and human observations to calculate the overlap (Figure 7). No analyses were possible for echidnas and lizards due to the small numbers observed. Long-nosed Bandicoot, possum and small mammal overlaps were high (Dhat 0.67, 0.68 and 0.67, respectively) with an overlap from dusk until dawn. The overlap of bird and human was low (Dhat 0.3 and 0.19, respectively), with most of the overlap occurring at dusk and dawn. One cat (Cat 7) was observed on camera 14 carrying something presumed to be a small- to medium-sized mammal—possibly a Long-nosed Bandicoot or rabbit—in its mouth (Figure 8).

## 4. Discussion

This study demonstrates the presence of multiple cats within critically important wildlife habitats on North Head, including within the Sydney Harbour National Park and North Head Sanctuary, where cats are prohibited. Several of the cats were observed wearing collars or were of identifiable breeds, and cats were most often detected using walking trails close to residential areas, suggesting that most, if not all, of the observed cats within the protected areas were owned pet cats roaming from adjacent suburban Manly. The time and location that the cats were observed overlapped with the activity of native animals known to be vulnerable to cat predation, such as Long-nosed Bandicoots, possums and other small mammals, with peaks in activity around dusk and in the early morning. While cats were the focus of this study, they were detected less frequently than foxes—another important introduced predator—and far less frequently than humans, and as such, their impact should be considered in proportion to the other interconnected threats to biodiversity on North Head.

Several cats were observed deep within the state and federally protected areas, including cats wearing collars, confirming that at least some pet cats roam long distances from home to access these areas. Of the five potential corridors cats might have used to access the protected areas of the headland, cats were more often detected using the walking trails with more vegetation cover over the corridors that were more exposed, especially the Collins Flat Track through littoral rainforest.

These findings of cat activity mainly at the boundary of the protected areas, are consistent with the literature on domestic cat roaming behavior, which has reported it is more common for cats that roam into natural areas to be found at the edges and less so within the centre of these natural areas [22]. Previous studies have also observed that cats are more likely to enter natural areas if there is cover [12,23]. Cats living closer to various types of natural areas in several settings, including in Australia, Great Britain, the United States and Norway have been noted to roam further and to have larger home ranges that extend into the natural area [12,13,14,15,24]. Domestic cats in Australia have also been noted to prefer travelling on or near established paths [25]. This might explain the multiple sightings of cats in centrally located areas of the North Head Sanctuary. The network of walking paths into and through this area likely facilitates cats having greater access to this sensitive wildlife habitat, exacerbating their potential impacts.

The owned domestic cats roaming within the state and federally protected areas are most likely those who live closest to the boundaries. Domestic cats can roam a considerable distance into natural habitats but rarely roam further than 1–2 km from home [22,23,26]. Barratt (1997) [27] found that cats would roam up to 900 m into the natural habitat, and Lilith et al. (2008) [28] also found that cats could move up to 300 m into bushland. Cats living near urban/natural edges are more likely to be seen within the natural boundaries than those living further from the edge [24,29]. Morgan et al. (2009) [29] reported that having a house closer to a natural wetland meant those cats were more frequent visitors that stayed longer than cats who live further away. López-Jara et al. (2021) [22] found that 63% of domestic cats using a conservation area lived within 100 metres of the area and that this distance has a direct association with cats roaming into the protected region. Lilith et al. (2008) [28] found at least 50% of rural pet cats whose homes bordered natural bushland prefer the bush areas. Pirie et al. (2022) [24] found cats living on a natural edge use those natural habitats up to 70% more than cats living away from the boundaries. They theorize that a cat’s ability to access these areas means they are more likely to use them [24]. Hence, living near the protected areas likely encourages owned domestic cats on North Head to roam further than they might otherwise.

Despite only recording one instance of possible cat predation, this study shows cats are frequently in the vicinity of threatened native animals, both inside the protected areas and in surrounding residential and parkland areas where Long-nosed Bandicoots were particularly common, and have overlapping activity, creating frequent opportunities for predation. Cats are opportunistic predators whose spatial and temporal movements vary considerably. Often, where a cat lives predicts how far and at what times they will roam [26,27,30]. The time of day cats roam is suggested to be dictated by factors including prey abundance [31]. The activity of the cats observed in this study had large peaks at 6 p.m. and 2 a.m., which overlapped with times when native mammals were most active. This was consistent with several studies that found that cats roam mostly during late evening or night [23,27,32,33]. This was consistent with Barratt (1997) [27], who reported that cats roaming at night roamed further and were more likely to roam into natural areas. This finding might also reflect cats adapting their behaviour to avoid people, as cats are prohibited within the national park, and human traffic is heavy during daylight hours. While cats vary widely in their interest in hunting and in their hunting success, cats do not distinguish between native and non-native prey species and domestic cats in urban areas can have a 28–52 times greater impact on wildlife per square kilometre than feral cats owing to their higher population density [11,34]. Pirie et al. (2022) [24] found that cats living near natural areas in Great Britain kill three times as many mammals as suburban cats. In addition, the presence of cats, including in urban settings, has been noted to affect wildlife physiology, behaviour, movements, space use and activity due to so-called “fear effects”, even without predation occurring [35,36,37].

The biodiversity of North Head is subject to multiple and complex environmental threats, of which cats are only one. As noted by Brook, Sodhi and Bradshaw (2008) [38], environmental threats can interact to have worse conservation outcomes than simply adding the individual impacts. Ongoing habitat destruction and fragmentation are the primary drivers of contemporary extinctions [38]. Meanwhile, the impacts associated with invasive predators have been recognised to exacerbate the impacts of land clearing, grazing and fire [39]. As generalist mesopredators, cats are better suited to fragmented habitats than larger predators and their impacts can be amplified in environments where larger predators (e.g., dingoes) are no longer present [40]. Resource subsidies by humans (i.e., well-fed pet cats) can also facilitate human-mediated hyperpredation; owned domestic cats are no longer subject to home-range restrictions and competition for territory due to hunger; hence, there is no limit to the number of cats they can “support” [39]. Predation and fear effects caused by cats across North Head cannot be considered in isolation. Their impacts compound the ongoing habitat loss and fragmentation occurring on the headland, as well as threats from high levels of human activity and the impacts of other invasive predators such as dogs and foxes. As such, it is important to understand how threats interact to understand the conservation risks and determine appropriate management strategies [41].

Understanding which sub-populations of cats are roaming within protected areas of North Head has important implications for designing interventions to reduce their potential impacts. Approaches to managing owned domestic cats—the cat subpopulation most likely represented in this study—are different from those required to manage unowned or feral populations [42]. Containment of pet cats to their caregiver’s property in NSW (the state where Manly is located) has steadily increased over several decades [43]. In addition, there is a growing evidence base on the design and implementation of cat caregiver behaviour change approaches to increase cat containment and reduce roaming [43,44,45]. However, evidence of their efficacy is limited. Previous research has indicated that individual cats vary considerably in their roaming and predation behavior [46,47]. Individual cats can also become specialist predators of a particular prey species and have been implicated in local extinctions [11]. As such, wildlife impacts associated with pet cats might not decrease without achieving very high rates of containment. Cats are already prohibited within Sydney Harbour National Park under the *National Parks and Wildlife Regulation 2019.* There are also prohibitions on cat and dog ownership imposed by the Sydney Harbour Trust on leasees of their properties at North Head. A similar restriction applies to the residents within the Spring Cove development to the North of Collins Beach. However, there are limited legislative provisions available to restrict cats from roaming within adjacent residential areas. As this study demonstrates, those residential areas also provide important habitat for endangered species, including the Long-nosed Bandicoot (*Perameles nasuta*). Increasing regulation of owned domestic cats roaming away from their caregiver’s property and increasing resourcing of enforcement of cat regulations might be required to reduce the roaming of cats observed in this study, especially to reduce the impacts associated with individual high-risk cats such as cats 4 and 7. Our findings suggest that enforcement activities would be most effective if they prioritized identifying the small number of individual high-impact cats and working with their caregivers to keep them contained. However, further research into the efficacy and potential unintended consequences of such regulation is urgently needed.

The reliance on camera traps in this study has associated limitations. The cameras most likely did not capture every cat event at each camera location, and the cameras only covered a small fraction of North Head. This study was only conducted over a 3-month period; a longer deployment or camera deployment at other times of the year might have yielded different results. In addition, all images were manually tagged, introducing the potential for human error. Further research that incorporates other techniques, such as transects, spotlighting and scat testing, would be beneficial. The use of tracking or video collars on owned cats could also be enlightening and might additionally influence cat caregiver behaviour change.

## 5. Conclusions

This study shows that cats are frequent visitors to the state and federally protected areas of North Head, and there is a clear temporal and spatial overlap between cats and native mammals who are at risk of cat predation. Several of the cats observed roaming within the national park were wearing collars or were of identifiable breeds and were most often observed on walking trails at the residential boundary, suggesting most, if not all, were owned domestic cats roaming from nearby suburban Manly. Endangered Long-nosed Bandicoots (*Perameles nasuta*) were also frequently observed in the residential and parkland locations where cats are legally allowed to roam. The findings of this study have important implications for approaches to reduce the impacts of cat predation. Interventions focused on cat caregiver awareness and behaviour change are indicated but might have limited success due to the disproportionate impacts of individual high-risk cats. Changes to cat regulation and enforcement might be required but exploration of their potential unintended consequences is urgently needed.

## Figures and Tables

**Figure 1 animals-14-02485-f001:**
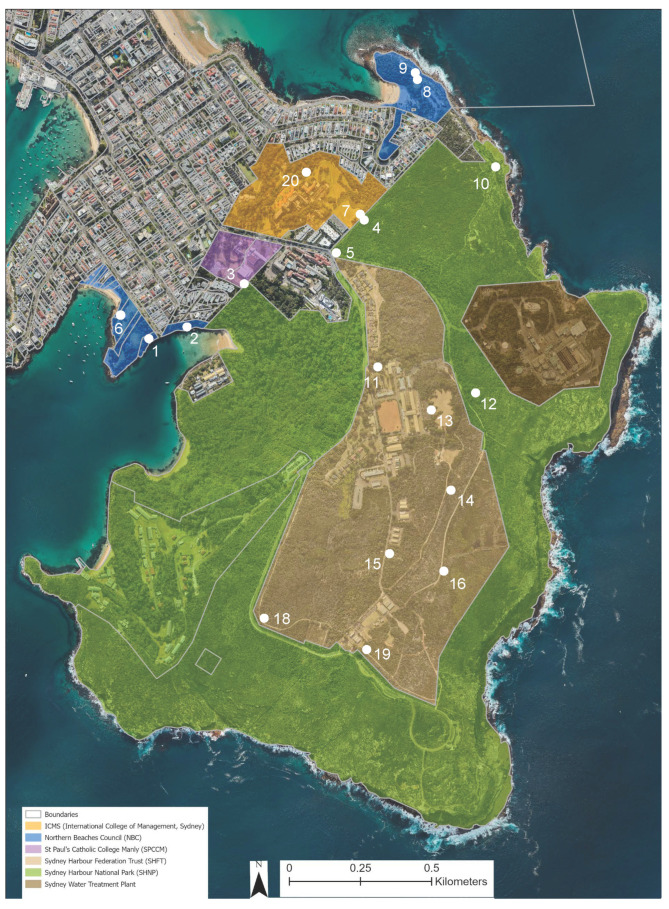
Map detailing the different land tenures at North Head, including the Sydney Harbour National Park and North Head Sanctuary (managed by the Sydney Harbour Federation Trust), where cats are prohibited. Numbered white dots indicate where monitoring cameras were located.

**Figure 2 animals-14-02485-f002:**
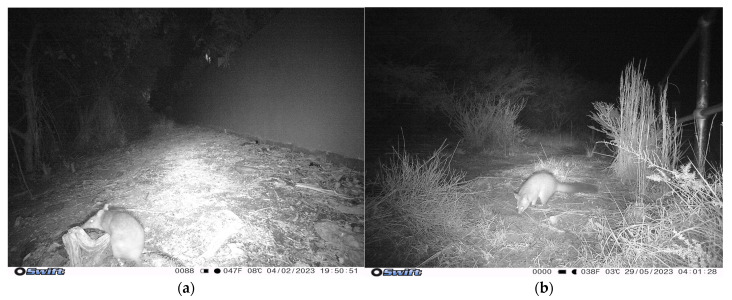
Images of wildlife: (**a**) Long-nosed Bandicoot (BAN) and (**b**) Brushtail Possum (POS) captured on camera at North Head, Manly.

**Figure 3 animals-14-02485-f003:**
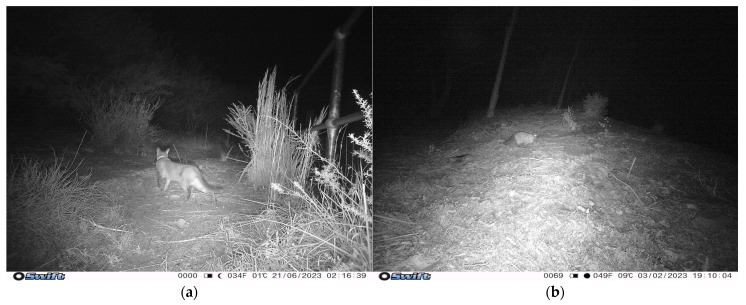
Images of cats observed on multiple cameras: (**a**) Cat 4 appeared in 14 events in 5 different locations, pictured here at camera 4 within the Sydney Harbour National Park; (**b**) cat 5 appeared on two cameras—camera 4 and camera 20 (pictured here—within the International College of Management, Manly, NSW, Australia.

**Figure 4 animals-14-02485-f004:**
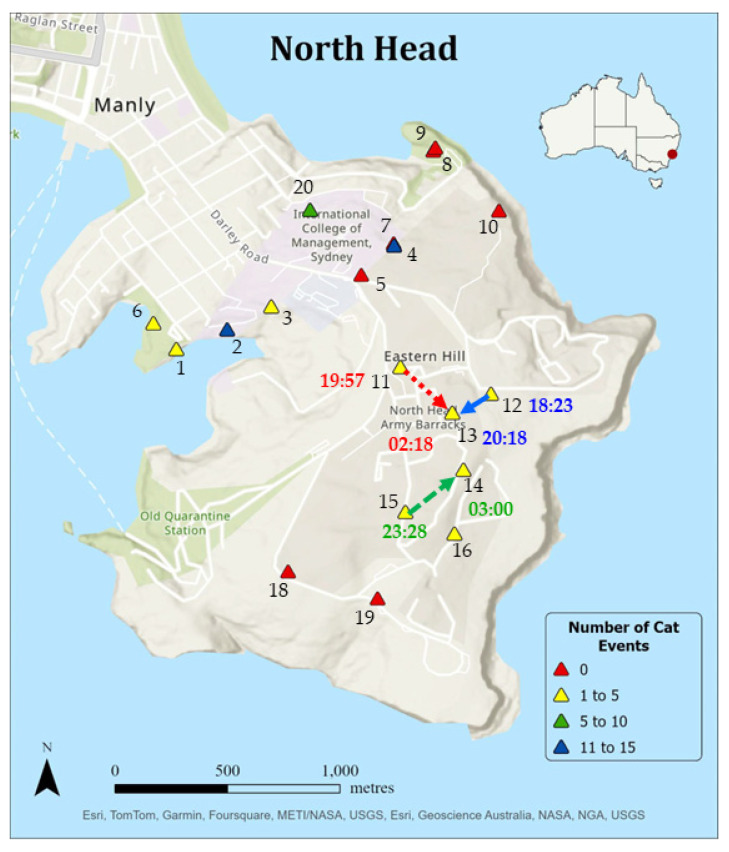
The number of cat events per camera along the boundary and within state and federal protected areas of North Head, NSW, Australia. Two cats were detected on more than one camera in a single night: Cat 4 was detected on two cameras on the one night on two occasions (dotted red and solid blue arrows), and cat 7 was detected twice in the one night once (dashed green arrow). The times of these detections are marked in the corresponding color.

**Figure 5 animals-14-02485-f005:**
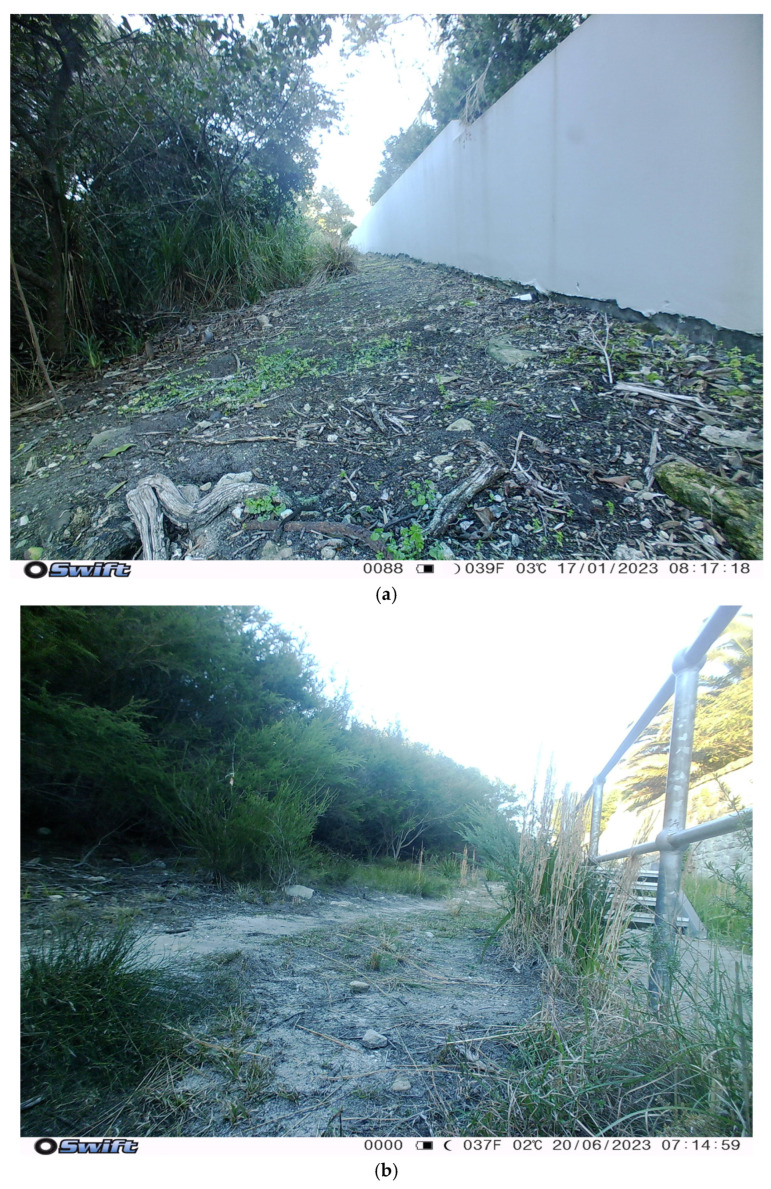
View from (**a**) camera 2 and (**b**) camera 4, which detected the most cat events of 19 cameras distributed across North Head, NSW, Australia.

**Figure 6 animals-14-02485-f006:**
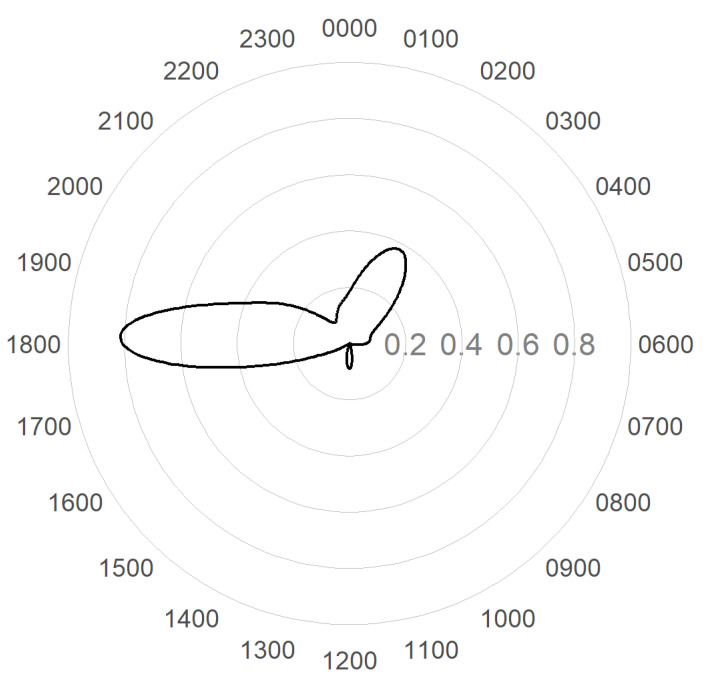
Diel plot showing the temporal movement of free-roaming cats at North Head, Manly, NSW, Australia.

**Figure 7 animals-14-02485-f007:**
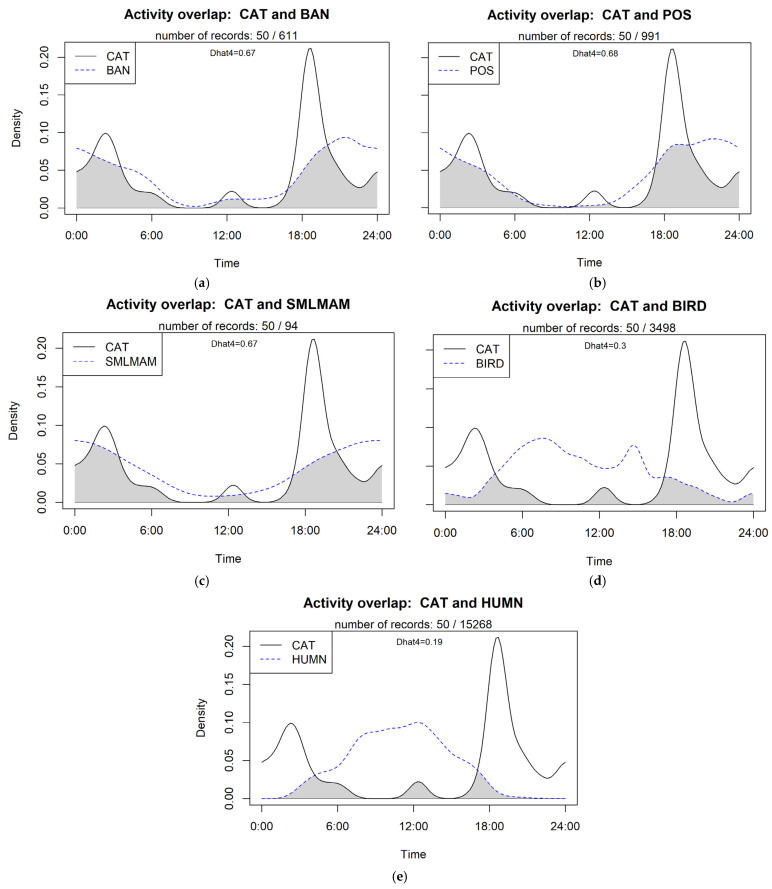
The temporal overlap of cats and (**a**) Long-nosed Bandicoots, (**b**) possums, (**c**) small mammals (**d**) birds and (**e**) humans at North Head, NSW, Australia.

**Figure 8 animals-14-02485-f008:**
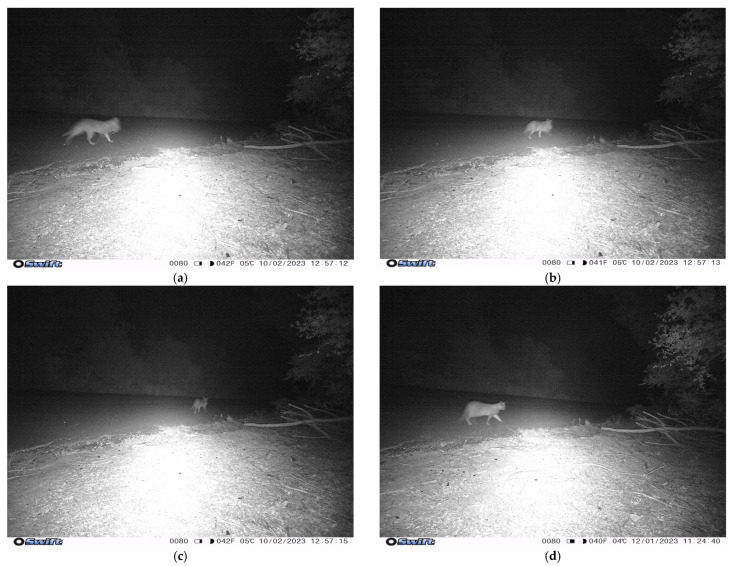
(**a**–**c**) Cat 7 with prey, presumed to be a Long-nosed Bandicoot or rabbit, captured on camera 14 at 1 a.m. within the North Head Sanctuary, Manly, NSW; and (**d**) the same cat two days prior in the same location without prey for comparison.

**Table 1 animals-14-02485-t001:** Weather conditions during the study period (May–July 2023). Source: Bureau of Meteorology.

Weather Conditions	May	June	July
Total Rainfall (mm)	43	19.6	35.4
Mean Min Temperature (°C)	12.1	11.6	11.1
Mean Max Temperature (°C)	19.4	18.1	18.7
Average time of sunrise	6:38 a.m.	6:56 a.m.	6:56 a.m.
Average time of sunset	5:04 p.m.	4:54 p.m.	5:05 p.m.

**Table 2 animals-14-02485-t002:** Number of events by species recorded on camera traps at North Head, Manly, NSW.

Camera	Land Tenure	BAN	BIRD	CAT	DDOG	ECH	FOX	HUMN	LIZ	POS	RAB	SMLMAM	Total	Trap Nights
1	NBC	123	354	1	13	1	0	21	0	40	87	8	648	56
2	NBC	34	219	15	0	0	0	2	0	47	29	20	366	56
3	SPCCM	119	129	2	0	1	0	2	1	3	584	1	842	56
4	SHNP	31	98	11	15	0	2	2914	0	268	134	2	3475	44
5	SHNP	4	56	0	14	0	0	2728	0	2	12	1	2817	41
6	NBC	1	60	2	8	0	0	44	0	3	21	0	139	56
7	ICMS	3	102	0	0	0	0	19	0	3	44	2	173	56
8	NBC	51	796	0	0	0	0	1	0	45	5	0	898	56
9	NBC	25	173	0	0	0	0	9	0	16	214	3	440	42
10	SHNP	0	1	0	1	0	0	1072	0	0	0	0	1074	8
11	SHFT	25	496	3	0	0	4	355	0	149	60	12	1104	56
12	SHNP	2	28	3	4	0	26	3297	0	110	126	2	3598	56
13	SHFT	37	80	2	3	0	18	292	0	103	680	5	1220	56
14	SHFT	37	12	3	3	0	4	884	0	72	158	4	1177	56
15	SHFT	24	21	1	0	0	3	632	1	9	14	0	705	56
16	SHFT	4	1	1	2	0	5	2855	0	77	11	0	2956	43
18	SHFT	2	2	0	0	0	0	0	0	0	2	12	18	56
19	SHFT	48	733	0	0	1	0	28	0	31	234	13	1088	56
20	ICMS	41	137	6	12	0	0	113	0	13	149	9	480	56
Total		611	3498	50	75	3	62	15,268	2	991	2564	94	23,218	963

‘NBC’ (Northern Beaches Council), ‘SPCCM’ (St Paul’s Catholic College, Manly), ‘SHNP’ (Sydney Harbour National Park), ‘ICMS’ (International College of Management Sydney), ‘SHFT’ (Sydney Harbour Federation Trust), ‘BAN’ (Long-nosed Bandicoot), ‘BIRD’, ‘CAT’, ‘DDOG’ (domestic dog), ‘ECH’ (echidna), ‘FOX’, ‘HUMN’ (human), ‘LIZ’ (lizard), ‘POS’ (possum), ‘RAB’ (rabbit/hare,) and ‘SMLMAM’ (other mammals smaller than a Long-nosed Bandicoot).

**Table 3 animals-14-02485-t003:** Individual cat events and the number of times each cat was observed per camera.

Camera	Land Tenure	Cats Prohibited?	Cat 1 # DSH	Cat 2 Unknown	Cat 3 DSH	Cat 4 # Burmese	Cat 5 # Ragdoll	Cat 6 DSH	Cat 7 Burmese	Cat 8 DSH	Cat 9 # Unknown	Cat 10 DSH	Cat 11 * Bengal	Unknown ^^^	Total
1	NBC	No						1							1
2	NBC	No	2	1	12										15
3	SPCCM	No										2			2
4	SHNP	Yes			2	6	2	1							11
5	SHNP	Yes													0
6	NBC	No											2		2
7	ICMS	No													0
8	NBC	No													0
9	NBC	No													0
10	SHNP	Yes													0
11	SHFT	Yes				3									3
12	SHNP	Yes				2								1	3
13	SHFT	Yes				2									2
14	SHFT	Yes							3						3
15	SHFT	Yes							1						1
16	SHFT	Yes				1									1
18	SHFT	Yes													0
19	SHFT	Yes													0
20	ICMS	No					1			4	1				6
Total			2	1	14	14	3	2	4	4	1	2	2	1	50

‘DSH’ (domestic shorthair), ‘NBC’ (Northern Beaches Council), ‘SPCCM’ (St Paul’s Catholic College, Manly), ‘SHNP’ (Sydney Harbour National Park), ‘ICMS’ (International College of Management Sydney), ‘SHFT’ (Sydney Harbour Federation Trust), # Cats wearing collars. * Cats controlled by humans, i.e., on a lead. ^^^ Cat image that was too indistinct to identify individually.

## Data Availability

All data are presented in this article.

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
