# Peer review of "Feline Encounters Down Under: Investigating the Activity of Cats and Native Wildlife at Sydney’s North Head"

_animals, 2024, doi:10.3390/ani14172485_

Round 1

Reviewer 1 Report

Comments and Suggestions for Authors

This paper adds to the literature demonstrating that cats, particularly owned cats, are a threat to indigenous wildlife. While the camera observations are helpful, I was disappointed that there were not more cameras, more deployments, and a longer time interval for monitoring. For example, you mention little penguins, but the cameras were not deployed during the breeding season? Also, the deployments locations did not seem well balanced to detect longer distance incursions (line 198,199). In general more effort should be expended in the potential areas of greatest concern, rather than at the border to maximise statistical power.

In terms of comments on the work done I have only one minor point about clarifying the possible presence of cats: Figure 1 test refers to "Sydney Harbour Sanctuary", but the figure shows "Sydney Harbour Federation Trust". Its not clear if these are the same thing or not. also clarify if there are no residential dwellings in the water treatment plant, so there should be no 'owned' cats here?

Author Response

COMMENT 1: While the camera observations are helpful, I was disappointed that there were not more cameras, more deployments, and a longer time interval for monitoring. For example, you mention little penguins, but the cameras were not deployed during the breeding season? Also, the deployments locations did not seem well balanced to detect longer distance incursions (line 198,199). In general more effort should be expended in the potential areas of greatest concern, rather than at the border to maximise statistical power.

RESPONSE 1: A longer deployment would have been interesting but was beyond the scope of this project. We have included this as a limitation in the Discussion as follows:

This study was only conducted over a 3-month period, a longer deployment, or camera deployment at other times of year might have yielded different results.”

COMMENT 2: In terms of comments on the work done I have only one minor point about clarifying the possible presence of cats: Figure 1 test refers to "Sydney Harbour Sanctuary", but the figure shows "Sydney Harbour Federation Trust". Its not clear if these are the same thing or not.

RESPONSE 2: We have replaced the map in Figure 1 to show the different land tenures more clearly. We have also clarified in the figure caption:

“Figure 1. a) Map showing the suburb of Manly, NSW, b) inset detailing the different land tenures on the Manly headland including the Sydney Harbour National Park and Sydney Harbour Sanctuary (managed by the Sydney Harbour Federation Trust) where cats are prohibited.”

And the Methods:

“The Manly headland includes of a variety of land tenures, including protected ar-eas where cats are prohibited (Figure 1). The Sydney Harbour National Park and Syd-ney Harbour Sanctuary (managed by the Sydney Harbour Federation Trust) are adja-cent to the residential area of Manly and include sclerophyll forest and littoral rain-forest remnants and coastal areas.”

COMMENT 3: Also clarify if there are no residential dwellings in the water treatment plant, so there should be no 'owned' cats here?

RESPONSE 3: We have clarified at line 110 that there are no residential dwellings within the Sydney Water Treatment Plan.

Reviewer 2 Report

Comments and Suggestions for Authors

ISSN 2076-2615 - Feline encounters Down Under: Investigating the activity of cats and native wildlife on Sydney’s Manly headland

General Comments

This paper described an estimate of the use of the Manly headland in Sydney, Australia by house cats. As domestic and feral cats exert established downward pressure on prey species populations, estimates of use of protected and wild areas by cats such as those presented here are important for wildlife management. However, this work could be improved by additional details in the methods and clarity in its goals and recommendations.

The state variables of interest are not clear in the introduction. Is the work attempting to estimate the population size of domestic cats, or just the cats’ activity patterns? Overlap with potential prey species?

Some additional details in the methods are needed to better understand how the photo tagging, summaries and analysis contributed to the study goals.

It is not clear what advance this work is making, beyond establishing that cats (feral or domestic) that live near the boundaries of restricted areas use those areas, a finding already established in the literature as this discussion section points out. An estimate of the level of predation or additional emphasis of the recommendations for mitigating cat use of protected areas listed at the end of the discussion would strengthen the utility of this work.

Inline comments

51-53: Please reference this sentence

82: Please specify what predation you are referring to. What prey species is implicated?

86: More detail is needed for the ‘halo effect’. A halo of reduced prey densities?

93: Please add more details into the study goals. What is meant by ‘subpopulations’? Is there a plan to estimate the actual population sizes of the species of interest?

105: Please clarify whether cats are prohibited from Sydney Harbour National Park and Sydney Harbour Sanctuary (beyond the figure caption 113)

153: Did events of the different species need to be separated by 60 seconds? If so, there could be missed events from important species.

168-175: Put the camera location description in the methods

181: How were individuals identified? Please report the confidence (error) in the IDs?

Author Response

COMMENT 1: The state variables of interest are not clear in the introduction. Is the work attempting to estimate the population size of domestic cats, or just the cats’ activity patterns? Overlap with potential prey species?

RESPONSE 1: The aims of the study have been clarified in the Introduction at line 94 as follows:

“This study aimed to determine the spatial and temporary activity of cats across the Manly headland. The study also aimed to define better understand which subpopula-tions of cats (feral, owned or unowned domestic) access protected areas of the Manly headland where cats are prohibited and determine which access points are usedto de-termine their temporal and spatial activity.”

COMMENT 2: Some additional details in the methods are needed to better understand how the photo tagging, summaries and analysis contributed to the study goals.

RESPONSE 2: This is addressed under the specific comments below.

COMMENT 3: It is not clear what advance this work is making, beyond establishing that cats (feral or domestic) that live near the boundaries of restricted areas use those areas, a finding already established in the literature as this discussion section points out. An estimate of the level of predation or additional emphasis of the recommendations for mitigating cat use of protected areas listed at the end of the discussion would strengthen the utility of this work.

RESPONSE 3: This comment contradicts comments made by Reviewer 3. As such, we have decided not to further adjust the discussion beyond changes made in response to the other two reviewers’ comments. An estimate of the level of predation is beyond the scope of this project, however we have made several recommendations in the Discussion for future studies, which include methods to better quantify predation.

Inline comments

COMMENT 4: 51-53: Please reference this sentence

RESPONSE 4: We have added a reference for this sentence:

            RSPCA Australia. Identifying best practice domestic cat management in Australia. 2018. https://kb.rspca.org.au/wp-content/uploads/2019/01/Identifying-Best-Practice-Domestic-Cat-Management-in-Australia-RSPCA-Research-Report-May-2018.pdf (accessed on 21 April 2023).

COMMENT 5: 82: Please specify what predation you are referring to. What prey species is implicated?

RESPONSE 5: This has been clarified as follows:

“While individual domestic cats are estimated to predate far fewer native animals than their feral counterparts, at a population level, predation pressure on native birds, mammals and reptiles exerted by domestic cats is estimated to be 30-50 times greater per square kilometer than predation by feral cats due to their greater population den-sity [11].”

COMMENT 6: 86: More detail is needed for the ‘halo effect’. A halo of reduced prey densities?

RESPONSE 6: This has been clarified as follows:

“As such, domestic cats are suspected to create a predation ‘halo effect’ where prey populations are reduced in the areas surrounding areas of urban development.”

COMMENT 7: 93: Please add more details into the study goals. What is meant by ‘subpopulations’? Is there a plan to estimate the actual population sizes of the species of interest?

RESPONSE 7: The aims of the study have been clarified as follows:

“This study aimed to determine the spatial and temporary activity of cats across the Manly headland. The study also aimed to define better understand which subpopula-tions of cats (feral, owned or unowned domestic) access protected areas of the Manly headland where cats are prohibited and determine which access points are usedto de-termine their temporal and spatial activity.”

COMMENT 8: 105: Please clarify whether cats are prohibited from Sydney Harbour National Park and Sydney Harbour Sanctuary (beyond the figure caption 113)

RESPONSE 8: This has been clarified as follows:

“The Manly headland includes of a variety of land tenures, including protected ar-eas where cats are prohibited (Figure 1). Cats are prohibited within the Sydney Har-bour National Park and Sydney Harbour Sanctuary (managed by the Sydney Harbour Federation Trust), which are adjacent to the residential area of Manly and include sclerophyll forest and littoral rainforest remnants and coastal areas.

COMMENT 9: 153: Did events of the different species need to be separated by 60 seconds? If so, there could be missed events from important species.

RESPONSE 9: There is no standard definition of an ‘event’ in the literature. Meek et al. (2014) states that, “a common camera-trapping term is an ‘event’, yet there are few useful definitions for this term in the literature, and no clear agreement about what an event means”. Our use of 60 seconds is consistent with other research monitoring free-roaming cats with cameras in our region.

Meek, P., Ballard, G., Claridge, A., Kays, R., Moseby, K., O’brien, T., O’Connell, A., Sanderson, J., Swann, D., & Tobler, M. (2014). Recommended guiding principles for reporting on camera trapping research. Biodiversity and Conservation, 23(9), 2321-2343.

COMMENT 10: 168-175: Put the camera location description in the methods

RESPONSE 10: This description has been moved to the Methods section in 2.2. Remote sensing cameras, and the camera locations have been added to Figure 1.

COMMENT 11: 181: How were individuals identified? Please report the confidence (error) in the IDs?

RESPONSE 11: This has been clarified at line 176 as follows:

“Cats were identified via their body markings by BK. Cats that were unable to be identified were listed as ‘Unknown’ and not counted as an individual in case the cat was captured elsewhere. Therefore, the number of individual cats presented is the minimum number of cats in the area at the time the research was conducted.”

Reviewer 3 Report

Comments and Suggestions for Authors

This study used camera traps to identify various species in and around nature areas in a tied island in Australia. They found many species, including cats, dogs, humans, and foxes in areas with threatened native species. Some also showed an overlap in time with these species. I’ve noted some additional options for intervention and some need to better situate the risk from cats in the greater risks from habitat destruction, fire, other human activity, etc. so as not to make all of the focus on cats when other important challenges will also need to be addressed for the welfare of native species. Specific comments below.

Intro: given the human habitat destruction, it seems like that should also be better woven into the first 2 paragraphs about cats or nuanced that cats are only one of the challenges for wildlife. And given the density of humans, seems like other types of human activities (bicycles, children) and dogs would also be problems. Given the large number of humans tagged, people themselves are of potential concern and cats are only one of many. This is mentioned but only within a paragraph in the discussion. Please edit in the intro and discussion and then the focus of this article can be the cats and their potential prey.

Line 82-83: this is stated in such a way as to imply that owned cats are individually worse than “feral” cats but really it is about population density. That is the lead here! Please edit text for clarity.

Line 84-85: Given the national variability in the #14  Hall reference (which is a survey and not cat-based data), it seems inconsistent to use all these references to support cat roaming in this way. Seems like a better argument would be to say that it is locally different and therefore this study was designed to evaluate this situation. The sentence that follows is appropriate, but this one isn’t.

Line 102-3: Reference 17: this is simply a number in a news article with no data or basis for its magnitude. Given the retrap of these cats, couldn’t an estimate be made? Even if not, it seems superfluous, especially since only 11 cats were seen in this study.

Nice explanation of the area, also good to see the climate data included and noted as typical!

Line 116: Please include average time of sunrise and sunset for these months for context.

Line 183: how was breed determined? Please reference in methods.

Line 187-8: I’m assuming that collar wearing, and breed are not overlapping? Please clarify a bit more.

Figure 4: please add to the legend that times are included and why only for those couple of spots…I’m assuming because the same cats were seen in two places on the same night?

Table 3: what does “Unknown” mean in this context? Please add to text and footnote.

Discussion: 11 seems like a fairly small number of cats for >900 trap nights. Especially since one was on leash. Throughout the discussion, please clarify which countries, level of ownership, and environments (urban, suburban, rural and how defined relative to current study) for the statements and references for context.

Line 258: again, these references aren’t in Australia. And are they likely owned or feral?

Line 264: which cats in this study are considered owned? Are some cats semi-owned/unowned domestic? Are any of these cats likely feral per the definitions in this manuscript? Please clarify in the results and in discussion. This isn’t included until late in the discussion.

Line 285-6: Cats also vary widely in their interest in hunting (as well as their success). Cats were mostly crepuscular, which is common in the literature, not all night long. Please be accurate in the text.

Line 298: this reference is from Great Brittain among owned cats. Does that apply here in this study, especially given the lack of predation visualized in all of these photos?

Line 302-3: As this study wasn’t about habitat fragmentation, only about cats and predation, this statement seems inappropriate.  Rather, this is fragmented habitat, and cats are only one of the many pressures including dogs and foxes.

Line 314-7: good, this is crucially important for this location and actually influences the ability of cats to hunt and the areas they hunt in as well.

Another approach to managing this location, is to identify the cats who spend the most time in the places they shouldn’t be in, trap them, place a paper collar on them with contact information for the owners, and then work with the owners to confine or restrict those specific cats. Clearly there are only a few cats (really cats 4 and 7, and maybe cats 3, 5, and 6) who are the main concern here. It could be useful to highlight that perhaps massive effort isn’t needed to protect this area from cats. Instead, if a few cats could be confined or contained during the night (recently shown to be a feasible human behavior change) a substantial decrease in the risk to wildlife AND the cats is more feasible. Since risk to wildlife isn’t a primary driver of human behavior change, perhaps risks to cats from dogs/foxes might be.

Line 350: radio collars or using video collars for owned cats in the boundary areas could also be instructive and the latter might be more useful in influencing cat owners to contain their cats for the cats’ welfare.

Author Response

COMMENT 1: Intro: given the human habitat destruction, it seems like that should also be better woven into the first 2 paragraphs about cats or nuanced that cats are only one of the challenges for wildlife. And given the density of humans, seems like other types of human activities (bicycles, children) and dogs would also be problems. Given the large number of humans tagged, people themselves are of potential concern and cats are only one of many. This is mentioned but only within a paragraph in the discussion. Please edit in the intro and discussion and then the focus of this article can be the cats and their potential prey.

RESPONSE 1: We thank Reviewer 3 for these recommendations to better contextualise our study with regards to other environmental threats. We have made some changes to the Introduction and extensive changes to the Discussion in response to the detailed comments as listed below.

COMMENT 2: Line 82-83: this is stated in such a way as to imply that owned cats are individually worse than “feral” cats but really it is about population density. That is the lead here! Please edit text for clarity.

RESPONSE 2: We have edited this sentence to distinguish between individual vs population level impcats as follows:

“While individual domestic cats are estimated to predate far fewer native animals than their feral counterparts, at a population level, predation pressure by domestic cats is estimated to be 30-50 times greater per square kilometer than predation by feral cats due to their greater population density [11].”

COMMENT 3: Line 84-85: Given the national variability in the #14  Hall reference (which is a survey and not cat-based data), it seems inconsistent to use all these references to support cat roaming in this way. Seems like a better argument would be to say that it is locally different and therefore this study was designed to evaluate this situation. The sentence that follows is appropriate, but this one isn’t.

RESPONSE 3: We have qualified this statement and have removed reference #14 and replaced with one that is more appropriate as follows:

“While cat roaming behaviour varies considerably, several studies in different settings have suggested that cats living adjacent to natural areas have also been shown to roam further and to prefer roaming within the natural areas [12-15].”

  1. Gehrt, S. D., Wilson, E. C., Brown, J. L., & Anchor, C. Population ecology of freeroaming cats and interference competition by coyotes in urban parks. PLoS One 2013, 8(9).

COMMENT 4: Line 102-3: Reference 17: this is simply a number in a news article with no data or basis for its magnitude. Given the retrap of these cats, couldn’t an estimate be made? Even if not, it seems superfluous, especially since only 11 cats were seen in this study.

RESPONSE 4: This statement has been deleted.

Nice explanation of the area, also good to see the climate data included and noted as typical!

COMMENT 5: Line 116: Please include average time of sunrise and sunset for these months for context.

RESPONSE 5: These averages have been included in Table 1 as follows:

Table 1. Weather conditions during the study period (May – July 2023). Source: Bureau of Meteorology.

Weather conditions

May

June

July

Total Rainfall (mm) 

43 

19.6 

35.4 

Mean Min Temperature (℃)

12.1 

11.6 

11.1 

Mean Max Temperature (℃)

19.4 

18.1 

18.7 

Average time of sunrise

6:38am

6:56am

6:56am

Average time of sunset

5:04pm

4:54pm

5:05pm

COMMENT 6: Line 183: how was breed determined? Please reference in methods.

RESPONSE 6: A description of how breed was determined has been included in the Methods section as follows:

“Using all the images tagged ‘CAT’, individual cats were identified and monitored across all cameras. Cats wearing collars, or of identifiable breeds were also noted. Breed was determined by a visual appraisal of images by GM.”

COMMENT 7: Line 187-8: I’m assuming that collar wearing, and breed are not overlapping? Please clarify a bit more.

RESPONSE 7: This data is provided in full in Table 3, but has also been clarified in the text as follows:

“Five individual cats, two of whom were wearing collars and were of an identifiable breed (Ragdoll and Burmese), and a third who was identified as a Burmese were detected by cameras located within state or federal protected areas where cats are prohibited in 24 separate events (Cats 3-7; Table 3).

COMMENT 8: Figure 4: please add to the legend that times are included and why only for those couple of spots…I’m assuming because the same cats were seen in two places on the same night?

RESPONSE 8: The legend has been edited as follows:

“Figure 4. The number of cat events per camera along the boundary and within state and federal protected areas of the Manly headland, NSW, Australia. Two cats were detected on more than one camera in a single night: Cat 4 was detected on two cameras on the one night on two occasions (dotted red and solid blue arrows), and Cat 7 was detected twice in the one night once (dashed green arrow). The times of these detections is marked in the corresponding color.”

COMMENT 9: Table 3: what does “Unknown” mean in this context? Please add to text and footnote.

RESPONSE 9: An explanation of this “unknown” cat has been added to the text as follows:

“Eleven individual cats were captured in 50 events on 63% of cameras (12/19; Table 3). One cat image detected on camera 12 was too indistinct to individually identify.”

An explanation has also been added to the caption of Table 3:

“‘DSH’ (Domestic Shorthair), ‘NBC’ (Northern Beaches Council), ‘SPCCM’ (St Paul’s Catholic College Manly), ‘SHNP’ (Sydney Harbour National Park), ‘ICMS’ (International College of Management Sydney), ‘SHFT’ (Sydney Harbour Federation Trust), # Cats wearing collars. *Cats controlled by humans i.e. on a lead. ^ Cat image that was too indistinct to indi-vidually identify.”

COMMENT 10: Discussion: 11 seems like a fairly small number of cats for >900 trap nights. Especially since one was on leash. Throughout the discussion, please clarify which countries, level of ownership, and environments (urban, suburban, rural and how defined relative to current study) for the statements and references for context.

RESPONSE 10: Thank you - this is a valid point. We have added additional qualification to the opening paragraph of the Discussion to put the findings of the study into context as follows:

“The time and locations cats were observed overlapped with the activity of native ani-mals known to be vulnerable to cat predation such as Long-nosed Bandicoots, possums and other small mammals, with peaks in activity around dusk and in the early morn-ing. While cats were the focus of this study, they were detected less frequently than foxes – another important introduced predator, and far less frequently than humans, and as such, their impact should be considered in proportion to other interconnected threats to biodiversity on the Manly headland.”

COMMENT 11: Line 258: again, these references aren’t in Australia. And are they likely owned or feral?

RESPONSE 11: These details have been included as follows:

“Cats living closer to various types of natural areas in several settings including in Australia, Great Britain, the United States and Norway have been noted to roam fur-ther, and to have larger home ranges that extend into the natural area [12-15,25]. Domestic cats in Australia have also been noted to prefer travelling on or near established paths [26].”

COMMENT 12: Line 264: which cats in this study are considered owned? Are some cats semi-owned/unowned domestic? Are any of these cats likely feral per the definitions in this manuscript? Please clarify in the results and in discussion. This isn’t included until late in the discussion.

RESPONSE 12:

An explanation to this effect has been included in the Results at line 194:

“All cats wearing collars and/or of identified breeds were considered owned domestic cats (6/11). The remaining five cats were all detected within 100m of human homes within suburban Manly (cameras 1-4, 20), hence these cats were also likely owned domestic cats, however it cannot be ruled out that these cats were unowned domestic cats.”

COMMENT 13: Line 285-6: Cats also vary widely in their interest in hunting (as well as their success). Cats were mostly crepuscular, which is common in the literature, not all night long. Please be accurate in the text.

RESPONSE 13: We have qualified statements around the times cats were detected to improve clarity at line 297:

“Despite only recording one instance of possible cat predation, this study shows cats are frequently in the vicinity of threatened native animals, both inside the protected areas and in surrounding residential and parkland areas (where Long-nosed Bandicoots were particularly common), and are active at the same times have overlapping activity, creating frequent opportunities for predation.”

And at line 302:

“The activity of cats observed in this study was strikingly nocturnal. Cats were mainly ob-served between 6.00pm and 2.00amhad large peaks at 6pm and 2am, which overlapped with times was also when native mammals were most active.”

And at line 309:

“While cats vary widely in their interest in hunting and in their hunting success, cats do not distinguish between native and non-native prey species and domestic cats in urban areas can have a 28-52 times greater impact on wildlife per square kilometer than feral cats owing to their higher population density [11,35].”

COMMENT 14: Line 298: this reference is from Great Brittain among owned cats. Does that apply here in this study, especially given the lack of predation visualized in all of these photos?

RESPONSE 14: Yes, we believe the findings of this British study do apply despite potential prey species being different as the species in question (Felis catus) is the same. We have clarified that this study was from Great Britain as follows:

“Pirie et al. (2022)[25] found cats living near natural areas in Great Britain kill three times as many mammals as suburban cats.”

COMMENT 15: Line 302-3: As this study wasn’t about habitat fragmentation, only about cats and predation, this statement seems inappropriate.  Rather, this is fragmented habitat, and cats are only one of the many pressures including dogs and foxes.

RESPONSE 15: We have revised the opening sentence of this paragraph to better contextualise the role of cats as follows:

“The biodiversity of the Manly headland provides an important example of how environmental threats such as habitat fragmentation are exacerbated by the presence of cats as an invasive predator is subject to multiple and complex environmental threats, of which cats are only one.”

COMMENT 16: Line 314-7: good, this is crucially important for this location and actually influences the ability of cats to hunt and the areas they hunt in as well.

RESPONSE 16: Thank you

COMMENT 17: Another approach to managing this location, is to identify the cats who spend the most time in the places they shouldn’t be in, trap them, place a paper collar on them with contact information for the owners, and then work with the owners to confine or restrict those specific cats. Clearly there are only a few cats (really cats 4 and 7, and maybe cats 3, 5, and 6) who are the main concern here. It could be useful to highlight that perhaps massive effort isn’t needed to protect this area from cats. Instead, if a few cats could be confined or contained during the night (recently shown to be a feasible human behavior change) a substantial decrease in the risk to wildlife AND the cats is more feasible. Since risk to wildlife isn’t a primary driver of human behavior change, perhaps risks to cats from dogs/foxes might be.

RESPONSE 17: Thank you for this suggestion. We have included your recommendation as follows:

“Increasing regulation of owned domestic cats roaming away from their caregiver’s property and increasing resourcing of enforcement of cat regulations might be re-quired to reduce the roaming of cats observed in this study, especially to reduce the impacts associated with individual high-risk cats such as Cats 4 and 7. Our findings suggest that enforcement activities would be most effective if they prioritized identi-fying the small number of individual high-impact cats, and working with their care-givers to keep them contained.”

COMMENT 18: Line 350: radio collars or using video collars for owned cats in the boundary areas could also be instructive and the latter might be more useful in influencing cat owners to contain their cats for the cats’ welfare.

RESPONSE 18: Thank you, this has been included at line 368:

“Further research that incorporates other techniques such as transects, spotlighting and scat testing would be beneficial. The use of tracking or video collars on owned cats could also be enlightening and might additionally influence cat care-giver behavior change.”